# Glycosylation Pathways Targeted by Deregulated miRNAs in Autism Spectrum Disorder

**DOI:** 10.3390/ijms26020783

**Published:** 2025-01-17

**Authors:** Federica Mirabella, Martina Randazzo, Alessandro Rinaldi, Fabio Pettinato, Renata Rizzo, Luisa Sturiale, Rita Barone

**Affiliations:** 1Child Neuropsychiatry Unit, Department of Clinical and Experimental Medicine, University of Catania, 95124 Catania, Italy; mirabella.federica.91@gmail.com (F.M.); randazzo96@gmail.com (M.R.); alessandrorinaldi96@libero.it (A.R.); fabiopettinato91@gmail.com (F.P.); rerizzo@unict.it (R.R.); 2CNR—Institute for Polymers, Composites and Biomaterials IPCB, 95126 Catania, Italy; luisella.sturiale@cnr.it; 3Research Unit of Rare Diseases and Neurodevelopmental Disorders, Oasi Research Institute—IRCCS, 94018 Troina, Italy

**Keywords:** miRNA, glycogenes, ASD, CDGs, biological pathways

## Abstract

Autism Spectrum Disorder (ASD) is a complex condition with a multifactorial aetiology including both genetic and epigenetic factors. MicroRNAs (miRNAs) play a role in ASD and may influence metabolic pathways. Glycosylation (the glycoconjugate synthesis pathway) is a necessary process for the optimal development of the central nervous system (CNS). Congenital Disorders of Glycosylation (CDGs) (CDGs) are linked to over 180 genes and are predominantly associated with neurodevelopmental disorders (NDDs) including ASD. From a literature search, we considered 64 miRNAs consistently deregulated in ASD patients (ASD-miRNAs). Computational tools, including DIANA-miRPath v3.0 and TarBase v8, were employed to investigate the potential involvement of ASD-miRNAs in glycosylation pathways. A regulatory network constructed through miRNet 2.0 revealed the involvement of these miRNAs in targeting genes linked to glycosylation. Protein functions were further validated through the Human Protein Atlas. A total of twenty-five ASD-miRNAs were identified, including nine miRNAs that were differentially expressed in cells or brain tissue in ASD patients and associated with glycosylation pathways, specifically protein N- and O-glycosylation and glycosaminoglycan biosynthesis (heparan sulfate). A number of CDG genes and/or ASD-risk genes, including *DOLK*, *GALNT2*, and *EXT1*, were identified as targets, along with validated interactions involving four key miRNAs (hsa-miR-423-5p, hsa-miR-30c-5p, hsa-miR-195-5p, and hsa-miR-132-5p). *B4GALT1*, an ASD susceptibility gene, emerged as a central regulatory hub, reinforcing the link between glycosylation and ASD. In sum, the evidence presented here supports the hypothesis that ASD-miRNAs mediate the epigenetic regulation of glycosylation, thus unveiling possible novel patho-mechanisms underlying ASD.

## 1. Introduction

Autism spectrum disorder (ASD) is a neurodevelopmental disturbance characterized by deficits in social communication and the occurrence of restricted interests and repetitive behaviors [1]. ASD affects one in thirty-six children in the United States (US) (Centers for Disease Control and Prevention, 2023) with a higher prevalence in males than in females. ASD biology is singularly challenging, including genetic and epigenetic contributions to the effects of multiple environmental factors [2]. Among epigenetic contributors to ASD, microRNAs (miRNAs) represent a group of small, non-coding RNA molecules that regulate gene expression at the post-transcriptional level and may have an abnormal expression in the body fluids and/or tissues of patients with ASD [3]. Deregulated miRNAs in ASD have been associated with the cell cycle, cell signaling, extracellular matrix–receptor interaction, and cancer, as well as with metabolism regulation [4]. ASD-miRNAs impact various metabolic pathways including steroid biosynthesis, biotin metabolism, fatty acid metabolism and biosynthesis, and lysine degradation [5]. Among metabolic pathways, glycosylation is a post-translational modification based on the attachment of carbohydrate chains (glycans) to proteins or lipids to synthesize glycoconjugates (glycoproteins, proteoglycans, mucins, glycolipids, and lipopolysaccharides). Glycan structures are extremely diverse, and their synthesis is regulated by genetic and environmental factors through a multitude of processing enzymes acting on specific sugar residues. These modifications are important for glycan’s final structure and function. Glycosylation influences pre- and postnatal brain development, synaptic plasticity, neurogenesis, neuronal migration, axon outgrowth and guidance, synaptogenesis, and neural plasticity [6]. Recent advances in the knowledge of neurogenesis have highlighted that cell polarity is important for morphological and functional diversity, and it is linked to glycosylation processes influencing brain development [7]. Unlike transcription and translation, glycosylation is not a template-driven process. As such, a variety of genetic, epigenetic, and environmental factors may influence glycosylation, resulting in an extraordinary diversity of glycan structures linked to health and pathological conditions. In such a context, post-translational modifications such as glycosylation have been linked to neurodevelopmental disorders (NDDs) and behavioral symptoms [8]. In general, NDDs constitute a broad group of conditions including global developmental delay, syndromic or nonsyndromic intellectual disability, ASD, attention deficit/hyperactivity disorder (ADHD), epilepsy, and motor disorders. It has emerged that the phenotypic outcomes of NDDs depend upon highly penetrant rare/de novo monogenic variants or common low-risk variants leading to multifactorial/polygenic disease. Much evidence suggests that the gains and losses of genes associated with glycosylation (*glycogenes*) might represent potential contributors to the development of NDDs such as ASD [8,9,10]. Serum glycome analyses of patients with ADHD have revealed recurrent modifications such as increased antennary fucosylation and decreased levels of some complex glycans with three or four antennas [11,12]. It has been postulated that decreased glycan branching might influence membrane receptors’ activity, thus representing a predisposal factor for ADHD [12]. Recently, it was found that single-nucleotide polymorphisms influencing serum immunoglobulin G (IgG) N-glycan traits have a causal relationship with psychiatric disorders by modulating the inflammatory response involved in immune-mediated neurological damage [13].

The unproper glycosylation of neuroligin 4 (NLN4) limiting protein surface trafficking has been associated with synaptic dysfunction and ASD [14]. In the context of syndromic ASD, we found a defective sialylation profile of cerebrospinal fluid in a patient with GM2 gangliosidosis indicating impaired N-glycosylation machinery in the Central Nervous System (CNS) associated with regressive ASD in the study patient [15]. Recently, a whole-transcriptome serum analysis of serum neural cell adhesion molecule L1 (L1CAM)-captured extracellular vesicles (LCEVs) showed that most RNAs differently expressed in ASD patients vs. controls involved neuron- and glycan-related networks supporting glycosylation changes in ASD [16].

The impact of glycosylation on CNS development and function is dramatically illustrated by a group of genetic diseases referred to as Congenital Disorders of Glycosylation (CDGs). CDGs are monogenic diseases caused by detrimental variants of glycogenes leading to defective glycan synthesis [17]. The majority of CDGs exhibit neurological symptoms such as hypotonia, ataxia, epileptic seizures, and stroke-like events, making glycosylation particularly crucial in the brain [18]. Patients with CDGs are affected with neurodevelopmental disturbances, mostly presenting with developmental delay/intellectual disability. ASD symptoms may occur in CDGs and may also represent hallmark features in CDG patients presenting with psychiatric symptoms [19]. Recent classification has accounted for 189 CDG-related gene defects resulting in almost 200 distinct phenotypes [20]. CDGs have been classified according to the affected glycosylation pathways, namely N-glycosylation, O-glycosylation, glycosylphosphatidylinositol (GPI)-anchor synthesis, lipid glycosylation, and other (including multiple) glycosylation pathways. The protein N- and O-glycosylation pathways consist of the covalent linkage of a sugar to the asparagine and the serine/threonine residues of a protein, respectively. The N-glycosylation pathway includes the multi-step biosynthesis of a dolichyl-pyrophosphate carrier of an oligosaccharide composed of fourteen glycan units (Dol-PP-GlcNAc2-Man9-Glc3). It is synthesized by the sequential addition of monosaccharide units requiring the action of several membrane-associated glycosyltransferases (GTs) that utilize nucleotide-activated sugars as donor substrates. This molecule is subsequently transferred “en-bloc” to the nascent glycoprotein and then processed to form a vast array of N-glycan structures. The second largest group of known CDGs affects the mannose O-glycosylation (O-Man) pathway causing phenotypes within the group of congenital muscular dystrophies [21]. Other CDGs are caused by deficiencies in glycogenes controlling different O-linked glycosylation pathways including O-Fuc, O-Glc, O-GlcNAc, O-Xyl, O-GalNAc, and HYL-Gal, as well as glycolipid and glycosylphosphatidylinositol (GPI) anchors [20].

The epigenetic regulation of glycosylation has been poorly described so far. Most of the evidence has focused on cancer, which has been associated with aberrant glycosylation associated with both genetic and epigenetic modifications of glycogenes [22]. Glycogenes with a role in normal neurodevelopment are epigenetically regulated: pivotal studies performed in the brain have found that the expression of glycogenes such as *MGAT5B*, *B4GALNT1*, and *ST8Sia1* is tightly controlled by histone modification [22].

miRNA epigenetic regulation in ASD has been extensively studied for both diagnostic purposes and understanding patho-mechanisms [23]. It has become clear that ASD-miRNAs may target various metabolic pathways pertinent to ASD biology. Recently, post-translation protein modifications such as glycosylation pathways have been increasingly linked to NDDs including ASD. However, information on the epigenetic regulation of glycosylation in ASD remains insufficient.

The present study aimed to understand whether and to what extent miRNAs, which are known to be deregulated in patients with ASD, may influence glycosylation.

For this purpose, we sought ASD-associated miRNAs with respect to their regulated pathways and validated target genes with special regards to glycogenes.

## 2. Results

### 2.1. Recognition of Differentially Expressed ASD-miRNAs

Of 64 studied miRNAs differently expressed and identified in at least two studies (ASD-miRNAs), we identified a subset of 25 ASD-miRNAs (39%) associated with glycosylation pathways and reported to be upregulated or downregulated in patients with ASD (Appendix A). Among these, we recognized 9 miRNAs that had been reported to be significantly differentially expressed (DE) in specific ASD cellular contexts compared with unaffected controls (Appendix A). In particular, hsa-miR-423-5p [24,25], hsa-miR-30c-5p [26,27], hsa-miR-195-5p [26,28,29], hsa-miR-199a-5p [30,31], hsa-miR-132-3p, and hsa-miR-132-5p [25,26,32,33] had been identified also in lymphoblastoid cell line (LCL). Moreover, we considered hsa-miR-379-5p [34,35], hsa-miR-21-3p [33,34], and hsa-miR-1277-3p [36,37], which had been identified in cerebral/cerebellar tissue.

### 2.2. Pathway Enrichment Analysis of Validated Targets for Each ASD-miRNA

Functional enrichment analyses were conducted to explore the potential biological involvement of ASD-miRNAs in molecular signaling pathways related to glycosylation processes. This investigation yielded a list of statistically over-represented biological pathways for each ASD-miRNA included in this study (Figure 1A,B). Among these, several glycosylation pathways were identified as being associated with one or more miRNAs (Figure 2): N-glycan biosynthesis (N = 2 miRNAs); glycosaminoglycan (GAG) biosynthesis–keratan sulfate (N = 3 miRNAs); mucin-type O-glycan biosynthesis (N = 3 miRNAs); glycosaminoglycan biosynthesis–chondroitin sulfate/dermatan sulfate (N = 1 miRNA); other types of O-glycan biosynthesis (N = 2 miRNAs); and glycosaminoglycan biosynthesis–heparan sulfate/heparin (N = 3 miRNAs). The assessment of study miRNAs in terms of their interaction with glycosylation pathways associated is reported in Table 1.

In addition, by merging the results through a pathway union (Figure 1A,B), from the most representative pathways, we identified metabolic processes such as fatty acid biosynthesis and metabolism (FDR corrected *p* < 1 × 10^−325^), glycosaminoglycan biosynthesis–keratan sulfate (FDR corrected *p* = 8.09 × 10^−11^), glycosaminoglycan biosynthesis–heparan sulfate/heparin (FDR corrected *p* = 0.0007), N-glycan biosynthesis (FDR corrected *p* = 0.0018), proteoglycans in cancer (FDR corrected *p* = 1.5 × 10^−10^), cell signaling pathways such as the Hippo signaling pathway (FDR corrected *p* = 2.8 × 10^−9^) and TGF-beta signaling pathway (FDR corrected *p* = 0.001), and lysine degradation (FDR corrected *p* = 1.5 × 10^−7^). These computational findings suggest that miRNAs identified as differentially expressed in ASD may play a functional role in molecular signaling pathways associated with glycosylation, which is particularly compelling.

### 2.3. Identification of Experimentally Validated Target Genes of Each miRNA

Based on the evaluation of miRNA–target interaction through TarBase v.8, we identified four ASD-miRNAs as the best-validated miRNAs that could strongly impact the regulation of glycosylation as follows: hsa-miR-423-5p (N-glycan biosynthesis, GAG biosynthesis–keratan sulfate), hsa-miR-30c-5p (mucin-type O-glycan biosynthesis), hsa-miR-195-5p (mucin-type O-glycan biosynthesis), and hsa-miR-132-5p (N-glycan biosynthesis, GAG biosynthesis–keratan sulfate, and other types of O-glycan biosynthesis) (Table 1).

This subset of miRNAs was particularly associated with N- and mucin-type O-glycosylation as well as with GAG biosynthesis and showed a good balance in terms of statistical significance, the prediction score of miRNA–target interaction, the number of publications supporting the target validation, and the number of glycogenes, including genes associated with human genetic glycosylation defects (CDG genes) and with ASD (ASD risk genes) as target genes (Figure 2).

### 2.4. Enrichment for Glycogenes and ASD Risk Genes

Interestingly, we found a significant number of CDG genes as part of ASD-miRNA target genes: *RPN1*, *DPM2*, *B4GALT1*, *ALG14*, *MAN1B1*, *DDOST*, *DOLK*, *MGAT1*, *ALG3*, *GALNT3*, *GALNT2*, *LFNG*, *POMT1*, *EXT1*, and *EXTL3* (Table 1). Simultaneously, among them, we identified ASD risk genes such as *DOLK*, *GALNT2*, and *EXT1*.

### 2.5. Reconstruction of the Whole miRNA-Mediated Regulatory Network

To explore the potential functional mechanisms of ASD-miRNAs, we constructed a regulatory network based on the nine miRNAs analyzed in this study.

Out of the most relevant gene nodes, we identified the glycogene *B4GALT1* with high values of connectivity and degree (Appendix A). Furthermore, in the supplementary functional analysis (Appendix A), we focused on ASD-miRNAs that are simultaneously implicated in specific glycosylation pathways to identify glycogenes that could serve as target gene nodes within the networks. Importantly, the following glycosylation pathways/glycogenes were found to be modulated by multiple ASD-miRNAs: N-glycosylation/*B4GALT1*; mucin-type O-glycosylation/*GALNT3* and *GALNT2;* O-glycosylation/*B4GALT1*, *OGT*, *LFNG*, and *POMT1;* and heparan sulfate biosynthesis/*EXT1* and *EXTL.*

### 2.6. Protein Function Analysis for Each Target Gene

Finally, protein function analysis highlighted that all target glycogenes identified through network analyses play essential roles linked to biological glycosylation processes, all encoding glycosyltransferase enzymes (*B4GALT1*, *GALNT3*, *GALNT2; OGT*, *LFNG*, *POMT1*, *EXT1*, *EXTL3*, and *LFNG*).

## 3. Discussion

Glycosylation influences pre- and postnatal brain development and synaptic plasticity, all processes that play a major role in neurodevelopment. The present study aimed to investigate the possible role of glycosylation as a target of epigenetic mechanisms mediated by miRNAs in NDDs with a focus on ASD, a condition characterized by validated miRNA deregulation [38,39]. Recently, miRNAs have gained prominence as important regulators of glycome [40]. However, the potential molecular influence of ASD-miRNAs on glycosylation processes has not been studied. Therefore, in this study, we first investigated the possible association between ASD-miRNAs and glycosylation pathways.

Our analysis of miRNA–target interaction showed that a subset of four ASD-miRNAs (hsa-miR-423-5p, hsa-miR-30c-5p, hsa-miR-195-5p, and hsa-miR-132-5p) was particularly related to glycosylation pathways including, as targets, a relevant number of OMIM genes associated with CDGs (CDG genes) and with ASD (ASD risk genes).

We identified nine CDG genes (i.e., *RPN1*, *DPM2*, *B4GALT1*, *ALG14*, *MAN1B1*, *DDOST*, *DOLK*, *MGAT1*, and *ALG3*) as targets of hsa-miR-423-5p.

Among these genes, *DOLK* mutations cause dolichol kinase deficiency, which affects the early steps of glycan biosynthesis. *DOLK*-CDG is characterized by epileptic encephalopathy, dysmorphic features, and variable systemic involvement. Notably, patients with *DOLK*-CDG may present with pure neurological phenotypes including ASD [41]. Moreover, *DOLK* variants are listed among ASD risk genes in patients with neurological disorders and ASD [42].

Among miR-30c-5p targets, *GALNT2* has a crucial role in healthy brain function [43]. *GALNT2* encodes the Golgi-localized polypeptide N-acetyl-d-galactosamine-transferase 2 isoenzyme, which acts in the initiation of mucin-type protein O-glycosylation. Interestingly, in patients, *GALNT2*’s loss of function causes a neurodevelopmental disorder with global developmental delay, intellectual disability with language deficit, autistic features, and behavioral abnormalities. Rodent models of *GALNT2*-CDG also have autistic-like behaviors with sensory processing and social impairment [43].

Furthermore, we found that *EXT1* (endoplasmic-reticulum-resident transmembrane glycosyltransferase), a significant “bottleneck” in the biosynthesis of heparan sulfate (HS), is targeted by ASD-miRNAs, miR-21-3p, miR-199a-5p, and miR-1277-3p. HS is a linear sugar formed by repeated units of N-acetylglucosamine and glucuronic acid, which links to proteins forming proteoglycan (HSPG) molecules. HSPGs are represented in the basal membrane and extracellular matrix, controlling cell migration, cell adhesion, and synaptogenesis. Several lines of evidence support a link between HS deficiency and ASD. Postmortem brain tissue has revealed reduced levels of HS in the lateral ventricles’ subventricular zone in individuals with ASD compared with age-matched neurotypical subjects [44]. Likewise, HS was found to be decreased in the same brain regions of the ASD mouse model BTBR T + tf/J, supporting the role of HS in interacting with growth factors and other ligands involved in brain development and ASD pathophysiology [45]. In the same context, it was found that knockout Ext1 mice displayed a deficiency of EXT in neurons and HS deficiency along with autistic behavior and abnormal glutamatergic transmission caused by reduced AMPA receptors’ synaptic expression [46]. In the clinical context, autosomal dominant mutations of *EXT1* causing familial exostosis were reported in two unrelated subjects with hereditary multiple exostoses, ASD, and intellectual disability [47]. Moreover, in the context of idiopathic ASD, a genome-wide association study (GWAS) meta-analysis including over 16,000 individuals with ASD definitely recognized *EXT1* as an ASD risk gene [48].

In the present study, we also identified *POMT1* and *LNFG* as target genes of miR-132-3p. *POMT1* encodes a glycosyltransferase, adding O-mannosyl glycans to dystroglycan (DG), whose deficiency causes muscular dystrophy and neural migration defects (Walker–Warburg syndrome). Individuals with this condition may have hydrocephaly, brain and retinal dysplasia, and significant cognitive difficulties that might lead back to ASD [49].

Alongside, *LFNG* plays a critical role in biological processes involving glycosylation because it encodes beta-1,3-N-acetylglucosaminyltransferase lunatic fringe, a member of the glycosyltransferase superfamily [50]. In one study, a patient with Asperger syndrome was reported with a de novo 380 kb gain in band p22.3 of chromosome 7 [50]. This duplicated region contains nine genes, including the *LFNG* gene. Our protein function analysis indicates that *LFNG* plays a pivotal role during embryonic development. It is a crucial regulator of NOTCH signaling, and its hyperactivation may result in an increase in neural progenitor cells at the expense of developing cells, leading to an expansion of brain structures [51,52]. It is noteworthy that ASD may be associated with larger head sizes and brain volumes during the first two years of life in a subset of patients [53].

Finally, the reconstruction of the regulatory network considering the nine miRNAs investigated in this study revealed that the *B4GALT1* glycogene exhibited the highest degree of connectivity, with a node degree of five. Therefore, *B4GALT1* emerged as a possible key regulatory gene node, which reinforces the link between glycosylation and ASD. *B4GALT1* encodes the β-1,4-galactosyltransferase 1 enzyme that participates in glycoconjugate biosynthesis, adding galactose to N-acetylglucosamine residues [54]. Several findings suggest that *B4GALT1* may be a susceptibility candidate gene related to glycobiology in ASD patients. It was identified in a copy number variation (CNV) screen associated with ASD [55]. A homozygous truncating mutation in *B4GALT1* was the cause of a non-complex ASD diagnosis in an individual with no other known genetic or environmental factors contributing to the condition.

In sum, we first described evidence indicating the role of ASD-miRNAs in regulating glycosylation, a fundamental biological process already implicated in ASD. We demonstrated that a number of CDG genes and/or ASD-risk genes, including B4GALT, DOLK, GALNT2, and EXT1, represent ASD-miRNAs targets, and we validated interactions involving four key miRNAs (hsa-miR-423-5p, hsa-miR-30c-5p, hsa-miR-195-5p, and hsa-miR-132-5p), thus unveiling possible novel patho-mechanisms underlying ASD.

## 4. Materials and Methods

### 4.1. Criteria for Considering Deregulated miRNAs in ASD for This Analysis

In the present study, we performed a literature search considering upregulated and downregulated miRNAs in patients with ASD. We considered clinical studies reporting (1) patients with ASD diagnosed by established diagnostic criteria and (2) miRNA expression changes that were significantly different in ASD patients compared to control subjects. The medical literature search was conducted on PubMed and Scopus from 2008 up to February 2023.

### 4.2. Selection of Studies

We divided miRNAs into up- and downregulated miRNAs in ASD, and we classified tissues and biological fluids where they were identified, such as in the cerebellar cortex, lymphoblastoid cell lines (LCLs), superior temporal sulcus (STS), primary auditory cortex (PAC), temporal lobe, frontal cortex (in particular, Brodmann areas 9 and 10 (BA9 and BA10)), temporal cortex (TC), saliva, blood, and serum. We highlighted ASD-associated miRNAs reported in two or more studies.

In the end, study miRNAs were selected according with the following inclusion criteria: miRNAs (1) reported to be deregulated in at least two independent studies (ASD-miRNAs) and (2) identified in a cellular context (i.e., lymphoblastoid cell line, frontal cortex, cerebellar cortex). Exclusion criteria were as follows: (1) pre-miRNAs, viral miRNAs, and non-miRNAs were not considered in the bioinformatic analysis, and (2) studies performed 15 or more years ago were not included.

### 4.3. Computational Analysis

#### 4.3.1. Computational Pathway Enrichment Analysis of Validated Targets for Each ASD-miRNAs

A computational pathway enrichment analysis was carried out to study the potential biological involvement of ASD-miRNAs on glycosylation processes using DIANA-miRPath v3.0 web server [56] based on KEGG (Kyoto Encyclopedia of Genes and Genomes) gene annotation database. DIANA-miRPath determines accurate functional characterization of one or more miRNAs using pathways and multiple gene ontologies. The KEGG analysis was used and the human species was selected. Fisher’s exact *t*-test (*p* ≤ 0.05) was applied for this analysis.

#### 4.3.2. Exploring the Experimentally Validated Target Genes of Each miRNA

All miRNA targets detected were analyzed by TarBase v.8 [57] to evaluate the experimental techniques used for the identification of miRNA–gene interactions, the prediction score, and the number of publications that supported the validation.

#### 4.3.3. Enrichment for Glycogenes and ASD Risk Genes Among Validated Targets

We computed the number of involved ASD-miRNAs identified for each glycosylation pathway and herein highlighted their molecular targets (glycogenes). Next, we considered if targeted glycogenes included OMIM-classified genes associated with CDGs [20]. Moreover, we evaluated whether targets of ASD-miRNAs in the glycosylation pathways were enriched for ASD risk genes reported in the Simons Foundation Autism Research Initiative and AutDB database (http://autism.mindspec.org/autdb/HG_Home.do (accessed on 21 June 2024)) including genes reported by GWAS studies, common variant association, genetic syndromes, and copy number variation [58].

#### 4.3.4. Reconstruction of miRNA-Mediated Regulatory Networks

Since miRNAs may act in concert, all miRNAs (N = 9) identified in this study were used to generate a miRNA-mediated regulatory network using miRNet 2.0 tool [59] to investigate the potential etiological role of miRNA-mediated network node genes. For this analysis, a cutoff of 1.0 was applied as a degree filter and performed on other nodes except miRNAs.

#### 4.3.5. Exploring KEGG Pathways That Are Commonly Targeted by Multiple miRNAs and Their Regulatory Networks

To evaluate the simultaneous involvement of multiple ASD-miRNAs acting to tune one individual glycosylation pathway and to identify their target genes, we extended the set of miRNAs (N = 25), performing extra functional analysis through the reconstruction of their regulatory networks.

#### 4.3.6. Exploring the Protein Function for Each Target Gene

For each glycogene nodes of all the regulatory networks reconstructed that showed the highest degrees, we explored the protein function and the related conditions using the Human Protein Atlas database [60].

## 5. Conclusions

The present study provided valuable insights into the potential link between ASD epigenetics, particularly miRNA deregulation, and glycosylation molecular mechanisms, a relationship that remains largely unexplored. Our findings indicate that deregulated ASD-associated miRNAs may be involved in the regulation of various glycosylation pathways, with a focus on protein N- and O-glycosylation and glycosaminoglycan biosynthesis, specifically heparan sulfate. Notably, the study identified ASD-miRNA targets that include glycogenes, whose detrimental mutations are linked to congenital disorders of glycosylation—metabolic conditions with significant neurological involvement, including ASD. Additionally, several ASD-miRNA targets overlap with ASD-risk glycogenes [3,61,62,63,64].

However, the study had limitations, including the need for validation in clinical samples from a larger and more phenotypically complete ASD patient population. Further experimental studies on miRNA expression profiles and their validation in larger patient cohorts are essential to elucidate the molecular basis of the potential association between glycosylation and ASD mechanisms. Beyond their diagnostic potential, these miRNAs could also serve as therapeutic targets as supported by the emerging literature [65,66].

In the end, this study demonstrated a robust correlation between NDDs, such as ASD, and neurometabolic diseases, thereby highlighting shared mechanisms that have the potential to inform advancements in diagnostic pathways and therapeutic interventions. Future research should prioritize the experimental validation of the identified miRNAs and their precise roles in glycosylation pathways, thus furthering their potential as both diagnostic biomarkers and therapeutic targets.

## Figures and Tables

**Figure 1 ijms-26-00783-f001:**
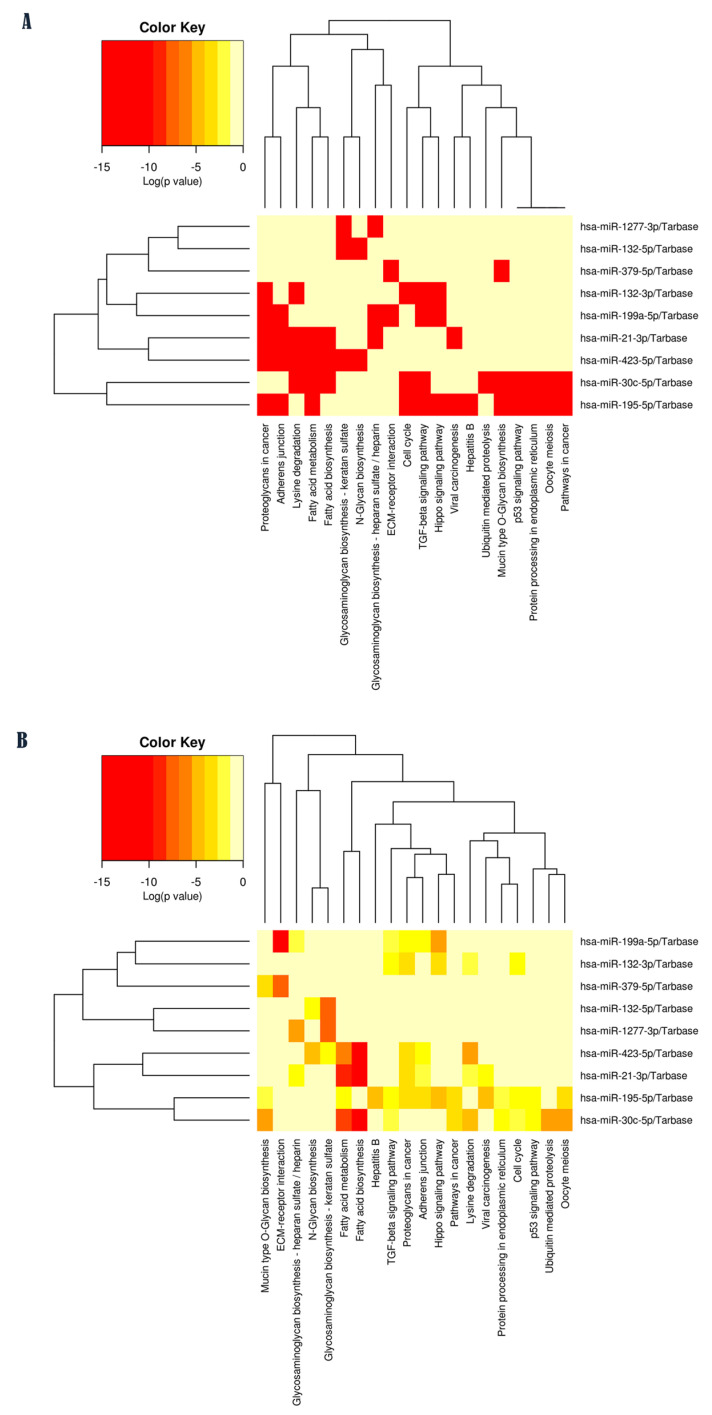
(**A**,**B**) Functional enrichment analysis of study ASD-miRNAs. (**A**) Targeted pathway clusters/heatmap illustrating results based on the existence (red) or absence (yellow) of statistical significance. (**B**) Significance clusters/heatmap illustrating results based on the effect sizes of statistical significance by using the exact significance levels of calculated *p*-values, represented with a gradient color (see legend). MiRNAs/KEGG pathway heatmaps were based on pathway union method to merge results and constructed by DIANA-mirPath v.3 web server. Dendrograms were designed using a complete linkage method with Euclidean distance measure and FDR corrected *p*-values were calculated by using Fisher’s exact test.

**Figure 2 ijms-26-00783-f002:**
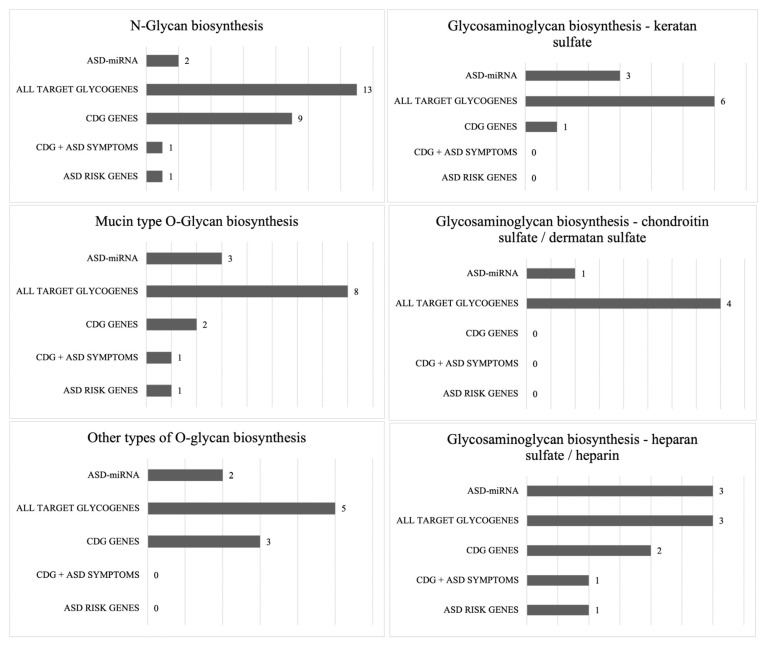
Glycosylation pathways and glycogenes targeted by study ASD-miRNAs.

**Table 1 ijms-26-00783-t001:** Study ASD-miRNAs’ interaction with glycosylation pathways: Evaluation of ASD-miRNAs with respect to their interaction with associated glycosylation pathways. Glycosylation pathways were identified from the KEGG database and significance levels (FDR corrected *p*-values) of miRNA–pathway interactions were calculated using Fisher’s exact test by miRPath v.3. The target genes of each glycosylation pathway and the prediction score of the interaction were found using TarBase v.8. N/A: not available.

ASD-miRNA	Glycosylation Pathways	FDR Corrected *p*-Value	Targeted Genes of Glycosylation Pathways	Prediction Score of Interaction	No. of Glycogenes	CDG Genes	ASD-Genes
hsa-miR-423-5p	N-glycan biosynthesis (hsa00510)	4.45 × 10^−5^	*RPN1*, *DPM2*, *B4GALT1*, *ALG10B*, *ALG14*, *MAN1B1*, *DDOST*, *DOLK*, *B4GALT3*, *MGAT1*, *ALG3*, *MGAT4B*	0.474 (*DPM2*) 0.661 (*ALG10B*)	9	*RPN1*, *DPM2*, *B4GALT1*, *ALG14*, *MAN1B1*, *DDOST*, *DOLK*, *MGAT1*, *ALG3*	*DOLK*
	Glycosaminoglycan biosynthesis–keratan sulfate (hsa00533)	5.82 × 10^−3^	*ST3GAL1*, *B4GALT1*, *B4GALT3*	N/A	1	*B4GALT1*	*/*
hsa-miR-30c-5p	Mucin-type O-glycan biosynthesis (hsa00512)	1.64 × 10^−6^	*GALNT7*, *B4GALT5*, *ST3GAL1*, *GCNT3*, *GALNT1*, *GALNT3*, *GALNT2*	1 (*GALNT7*) 0.559 (*B4GALT5*) 0.507 (*GCNT3*) 0.999 (*GALNT1*) 0.992 (*GALNT3*) 0.996 (*GALNT2*)	2	*GALNT3*, *GALNT2*	*GALNT2*
hsa-miR-195-5p	Glycosaminoglycan biosynthesis–chondroitin sulfate/dermatan sulfate (hsa00532)	1.62 × 10^−2^	*UST*, *DSE*, *CHPF*, *CHPF2*	0.457 (*UST*) 0.564 (*CHPF*)	0	*/*	*/*
	Mucin-type O-glycan biosynthesis (hsa00512)	3.87 × 10^−2^	*GALNT7*, *GALNT1*, *GALNT3*, *GALNT2*	0.784 (*GALNT7*) 0.659 (*GALNT1*)	2	*GALNT3*, *GALNT2*	*GALNT2*
hsa-miR-132-3p	Other types of O-glycan biosynthesis (hsa00514)	2.69 × 10^−2^	*LFNG*, *B3GAT1*, *POMT1*, *EOGT*	N/A	2	*LFNG*, *POMT1*	*/*
hsa-miR-21-3p	Glycosaminoglycan biosynthesis–heparan sulfate/heparin (hsa00534)	5.14 × 10^−3^	*EXT1*, *EXTL3*, *NDST2*	0.503 (*EXT1*)	2	*EXT1*, *EXTL3*	*EXT1*
hsa-miR-132-5p	Glycosaminoglycan biosynthesis–keratan sulfate (hsa00533)	1.85 × 10^−8^	*B4GALT1*, *B3GNT1*	0.537 (*B4GALT1*)	1	*B4GALT1*	*/*
	N-glycan biosynthesis (hsa00510)	6.53 × 10^−3^	*GANAB*, *B4GALT1*	0.537 (*B4GALT1*)	1	*B4GALT1*	*/*
	Other types of O-glycan biosynthesis (hsa00514)	9.66 × 10^−3^	*B4GALT1*	0.537 (*B4GALT1*)	1	*B4GALT1*	*/*
hsa-miR-199a-5p	Glycosaminoglycan biosynthesis–heparan sulfate/heparin (hsa00534)	4.90 × 10^−2^	*EXT1*	0.67 (*EXT1*)	1	*EXT1*	*EXT1*
hsa-miR-1277-3p	Glycosaminoglycan biosynthesis–keratan sulfate (hsa00533)	1.06 × 10^−8^	*CHST1*, *B3GNT2*	N/A	0	*/*	*/*
	Glycosaminoglycan biosynthesis–heparan sulfate/heparin (hsa00534)	1.66 × 10^−6^	*EXT1*	N/A	1	*EXT1*	*EXT1*
hsa-miR-379-5p	Mucin-type O-glycan biosynthesis (hsa00512)	6.53 × 10^−4^	*GALNT11*	N/A	0	*/*	*/*

## Data Availability

The datasets generated during and/or analyzed during the current study are available from the corresponding author on reasonable request.

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
