# Peer review of "Glycosylation Pathways Targeted by Deregulated miRNAs in Autism Spectrum Disorder"

_ijms, 2025, doi:10.3390/ijms26020783_

Round 1
Reviewer 1 Report
Comments and Suggestions for Authors
To the Authors
MicroRNAs (miRNAs) are known to play a role in ASD and may influence metabolic pathways. In particular, glycosylation is necessary for the optimal development of the central nervous system (CNS). Genetic defects of glycosylation (CDGs) are linked to over 180 genes and are predominantly associated with neurodevelopmental disorders, including ASD. By using computational tools, the Authors have investigated the potential involvement of ASD-miRNAs (ASD-miRNAs) in glycosylation pathways after selecting from the literature 64 miRNAs consistently deregulated in ASD. In addition, the Authors have validated the corresponding protein function by the Human Protein Atlas. A total of 25 ASD-miRNAs have been identified, including nine miRNAs that were differentially expressed in cells or brain tissue of ASD patients and associated with glycosylation pathway. A number of CDG-genes and/or autism-risk genes, including DOLK, GALNT2, and EXT1, have been identified as targets, and interactions involving four key miRNAs (hsa-miR-423-5p, hsa-miR-30c-5p, hsa-miR-195-5p and hsa-miR-132-5p) were validated by the Authors. Specifically, B4GALT1, an ASD susceptibility gene, emerged as a central regulatory hub, reinforcing the link between glycosylation and ASD. Although the topic is certainly of interest, several issues regarding data presentation and novelty should be carefully addressed.
Major Issues
1. The elements of novelty of the current study should be better emphasized.
2. Discussion should be consistently rephrased. A) I would suggest a shorter incipit of the Discussion, limiting it to one or two single introductory sentences; B) I would suggest avoiding repeating results in the Discussion. On the opposite, the Authors are encouraged to concentrate more on the discussion and interpretation their results
3. Although The quality of English does not limit the understanding of the research, I would suggest improving the quality of language style.
Minor Issues
4. The following sentence in the Discussion has no apparent grammatical sense and should be rephrased: “Finally, the reconstruction of the entire miRNA-mediated regulatory network, vali-dated B4GALT1 as having the higher value of degree and betweenness supporting a crucial role in study ASD miRNA-mediated regulatory network”.
5. In the hierarchical clusters and miR-Nas/KEGG pathways heatmap there are no color nuances. Is that implying that log(p value) is not applied? What is the statistical significance linked to either up- or down-regulation? Please clarify.
6. The p-value reported in Table 1 is relative to the association between mRNA and the target gene rather than the comparison of expression between ASD individuals and normotypic controls. Please clarify.
7. Although the references are appropriate, just above one quarter of the citations are dating in the last 5 years (2020-2024: 27.8%). I would encourage the Authors to further update their reference list.
Author Response
Reviewer 1
We thank the Reviewer for providing us with suggestions and comments.
MicroRNAs (miRNAs) are known to play a role in ASD and may influence metabolic pathways. In
particular, glycosylation is necessary for the optimal development of the central nervous system
(CNS). Genetic defects of glycosylation (CDGs) are linked to over 180 genes and are predominantly
associated with neurodevelopmental disorders, including ASD. By using computational tools, the
Authors have investigated the potential involvement of ASD-miRNAs (ASD-miRNAs) in
glycosylation pathways after selecting from the literature 64 miRNAs consistently deregulated in
ASD. In addition, the Authors have validated the corresponding protein function by the Human
Protein Atlas. A total of 25 ASD-miRNAs have been identified, including nine miRNAs that were
differentially expressed in cells or brain tissue of ASD patients and associated with glycosylation
pathway. A number of CDG-genes and/or autism-risk genes, including DOLK, GALNT2, and EXT1,
have been identified as targets, and interactions involving four key miRNAs (hsa-miR-423-5p, hsamiR-30c-5p, hsa-miR-195-5p and hsa-miR-132-5p) were validated by the Authors.
Specifically, B4GALT1, an ASD susceptibility gene, emerged as a central regulatory hub, reinforcing
the link between glycosylation and ASD. Although the topic is certainly of interest, several issues
regarding data presentation and novelty should be carefully addressed.
Major Issues
1. The elements of novelty of the current study should be better emphasized.
Thank you for the comment. The novelty of the study was indicated in proper paragraphs
(introduction, discussion and conclusion)
2.Discussion should be consistently rephrased. A) I would suggest a shorter incipit of the
Discussion, limiting it to one or two single introductory sentences;
I would suggest avoiding repeating results in the Discussion.
On the opposite, the Authors are encouraged to concentrate more on the discussion and
interpretation their results
Thank you for your valuable comment: the discussion has been shortened using a shorter
incipit to the topic. It was more focused on the main results of the study.
3.Although The quality of English does not limit the understanding of the research, I would
suggest improving the quality of language style.
Thanks for suggestions: language style was revised.
Minor Issues
4.The following sentence in the Discussion has no apparent grammatical sense and should be
rephrased: “Finally, the reconstruction of the entire miRNA-mediated regulatory network,
vali-dated B4GALT1 as having the higher value of degree and betweenness supporting a
crucial role in study ASD miRNA-mediated regulatory network”.
Thank you for your correct remark. As you suggested, we clarified the following information:
“Finally, the reconstruction of the regulatory network considering the nine miRNAs investigated in
this study, revealed that the B4GALT1 glycogene exhibited the highest degree of connectivity, with a
node degree of five. Therefore, B4GALT1 emerged as a possible key regulatory gene-node which
reinforces the link between glycosylation and ASD.”.
5.In the hierarchical clusters and miR-Nas/KEGG pathways heatmap there are no color
nuances. Is that implying that log(p value) is not applied? What is the statistical significance
linked to either up- or down-regulation? Please clarify.
Thank you for your observation. Figure 1, representing hierarchical clusters and
miRNAs/KEGG pathways heatmap, has been constructed by miRPath web server by selecting
the option “Targeted Pathways Clusters/Heatmap”. This possibility offers to flag with 0 all
the significantly targeted pathways (showing p-value under the threshold, i.e. 0.05), and 1
otherwise. This selection is based on the existence/absence of statistical significance. We
decided to display this type of heatmap to make the observed significance clearer and more
immediate for the readers.
Nevertheless, to clarify the result as suggested, in the revised version of the manuscript we
have updated the figure caption and the respective paragraph (2.2) with more relevant
information and added a new figure (Figure 1-B). This illustration has been constructed by
selecting the “Significance Clusters/Heatmap” option, that utilizes the exact significance
levels calculated by Fisher’s exact test representing results based on the effect size.
6. The p-value reported in Table 1 is relative to the association between mRNA and the
target gene rather than the comparison of expression between ASD individuals and
normotypic controls. Please clarify.
We clarified the significance of p-values reported in Table 1, by adding more information in
the caption as following “Table 1. Study ASD-miRNAs interaction with glycosylation pathways.
Evaluation of ASD-miRNAs with respect to their interaction with associated glycosylation pathways.
Glycosylation pathways were identified from the KEGG database and significance levels (FDR
corrected p-values) of miRNA-pathway interactions were calculated using Fisher's exact test by
miRPath v.3. The target genes of each glycosylation pathway and the prediction score of the
interaction were performed using TarBase v.8.
7.Although the references are appropriate, just above one quarter of the citations are dating in
the last 5 years (2020-2024: 27.8%). I would encourage the Authors to further update their
reference list.
Thank you for your annotation: reference list was updated whenever possible

Reviewer 2 Report
Comments and Suggestions for Authors
The manuscript title „Glycosylation pathways targeted by deregulated miRNas in Autism Spectrum Disorder (ASD)” gives an overview of the proposed subject.
Within the introduction section, it is of interest to detail the subject of glycosylation pathways involved having in view the relevance of the ASD prevalence. Having in view the fact that neurodevelopmental disorders are key points of the paper, it is important to add some information regarding the involvment of glycosylation for neurodevelopmental disorders.
The section of the results includes a large amount of data on identified ASD-miRNAs and their association with glycosylation pathways. The authors should improve the text with information that correlate with key findings from the table.
The study’s limitations should also be discussed as well as the presentation of the involvment of the current findings for future research or clinical practice as targets of therapeutic interventions.
The conclusion highlights the findings and could be improved with information regarding the wider ramifications of the research. It should also propose specific domains for future investigation, including the necessity for experimental validation of the discovered miRNAs and their functions in glycosylation pathways.
Author Response
Reviewer 2
We thank the Reviewer for all comments and suggestions.
The manuscript title „Glycosylation pathways targeted by deregulated miRNas in Autism Spectrum
Disorder (ASD)” gives an overview of the proposed subject.
Within the introduction section, it is of interest to detail the subject of glycosylation pathways
involved having in view the relevance of the ASD prevalence. Having in view the fact that
neurodevelopmental disorders are key points of the paper, it is important to add some information
regarding the involvment of glycosylation for neurodevelopmental disorders.
Thank you for your consideration. We implemented the introduction session better explaining the
possible involvement of glycosylation changes in neurodevelopmental disorders and adding
information about serum glycome changes already found in patients with neurodevelopmental
disorders.
The section of the results includes a large amount of data on identified ASD-miRNAs and their
association with glycosylation pathways. The authors should improve the text with information that
correlate with key findings from the table.
Thank you for your suggestion. As suggested, in the revised manuscript we focused on fundamental
results provided in Table 1 and we moved additional results on supplementary functional analysis,
(previous paragraph 2.6) in the supplementary material
The study’s limitations should also be discussed as well as the presentation of the involvment of the
current findings for future research or clinical practice as targets of therapeutic interventions. The
conclusion highlights the findings and could be improved with information regarding the wider
ramifications of the research.
We agree with the Reviewer’s comment. Limitations of the study have been reported and the possible
perspectives of the current research were also discussed in the conclusion paragraph.
It should also propose specific domains for future investigation, including the necessity for
experimental validation of the discovered miRNAs and their functions in glycosylation pathways.
Thank you for your suggestions. This point was underlined in the discussion and conclusion
sections.

Round 2
Reviewer 1 Report
Comments and Suggestions for Authors
To the Authors
The Authors have carefully and thoughtfully addressed all the raised points of constructive criticism, and the Ms accuracy is improved both in terms of readability and understanding by the ASD research community.